# The Apelinergic System in Pregnancy

**DOI:** 10.3390/ijms24098014

**Published:** 2023-04-28

**Authors:** Océane Pécheux, Ana Correia-Branco, Marie Cohen, Begoῆa Martinez de Tejada

**Affiliations:** 1Obstetrics Division, Department of Woman, Child and Adolescent, Geneva University Hospitals, 1205 Geneva, Switzerland; 2Department of Pediatrics, Gynecology and Obstetrics, Faculty of Medicine, University of Geneva, 1205 Geneva, Switzerland

**Keywords:** apelin, Elabela, APJ, placenta, pregnancy, preeclampsia

## Abstract

The apelinergic system is a highly conserved pleiotropic system. It comprises the apelin receptor apelin peptide jejunum (APJ) and its two peptide ligands, Elabela/Toddler (ELA) and apelin, which have different spatiotemporal localizations. This system has been implicated in the regulation of the adipoinsular axis, in cardiovascular and central nervous systems, in carcinogenesis, and in pregnancy in humans. During pregnancy, the apelinergic system is essential for embryo cardiogenesis and vasculogenesis and for placental development and function. It may also play a role in the initiation of labor. The apelinergic system seems to be involved in the development of placenta-related pregnancy complications, such as preeclampsia (PE) and intrauterine growth restriction, but an improvement in PE-like symptoms and birth weight has been described in murine models after the exogenous administration of apelin or ELA. Although the expression of ELA, apelin, and APJ is altered in human PE placenta, data related to their circulating levels are inconsistent. This article reviews current knowledge about the roles of the apelinergic system in pregnancy and its pathophysiological roles in placenta-related complications in pregnancy. We also discuss the challenges in translating the actors of the apelinergic system into a marker or target for therapeutic interventions in obstetrics.

## 1. Overview of the Apelinergic System

The apelinergic system is composed of a group of three actors, namely, a receptor named apelin peptide jejunum (APJ) and its two peptide ligands, Elabela/Toddler (ELA) and apelin [1]. The APJ gene, APLNR, was discovered in 1993 and showed homology with the angiotensin II type 1 receptor [1,2]. However, APJ, a seven-transmembrane G protein-coupled receptor (GPCR), did not bind to angiotensin II [2] and was initially considered as an orphan GPCR [1,2]. Its first endogenous ligand, the peptide hormone apelin, was discovered several years later in 1998 by Tatemoto et al. by means of monitoring APJ activity from bovine stomach extracts [3].

APJ and the preproapelin, consisting of 77 amino acid residues, are expressed in embryo and adult human tissues, including heart, vasculature (particularly in endothelial cells), and lung tissue; white adipose tissue; the gastrointestinal tract and the liver; several regions of the central nervous system; retinas; limbs; the skin; kidneys; mammary glands; and placental tissue [1,4,5,6,7,8,9,10,11,12,13,14]. The preproapelin can be cleaved from its C-terminal domain to produce several apelin peptides with different polypeptide chain lengths (apelin-36, apelin-17, and apelin-13). Research has shown that the longer chains of this protein are characterized by lower biological activity, which is why they are converted into short-chain forms [15]. Apelin-36 predominates in rat lung, testis, and uterus [16] and in bovine colostrum [3]. Its concentration is much lower in rat brain as well as in rat and human plasma, where the most abundant forms of apelin are apelin-17 and pyroglutamate-apelin-13 [17,18]. The naturally pyroglutamated apelin-13 form is structurally more resistant to aminopeptidases and is also the most active isoform. It is located in the mammary gland and hypothalamus [16], but also in the heart, where it is the most abundant form [19].

A second endogenous ligand, ELA, was identified in 2013 in zebrafish embryos [20,21] by Chng et al. While seeking to identify the first hormonal peptide implicated in the ability of naive blastomeres to differentiate into one of the three embryonic germ layers, they isolated a human gene named ‘APELA’ (apelin early endogenous ligand), annotated until then as a noncoding transcript. APELA was predicted to encode a hormone with a signal peptide, ELA [20]. Concurrently, Pauli et al. also identified the same gene and named it ‘TODDLER’ [21]. Thus, even if they both bind to APJ, ELA and apelin differ not only in their structure [22] but also by their encoding genes, which is rather unusual for peptide ligands of the same GPCR. ELA is the early ligand in humans, but it remains present in blood during adulthood by means of its expression in the prostate, the kidney, the cardiac endothelium, blood vessels, and the placenta [20,23,24,25,26,27]. Its crucial role in early human development will be further reviewed in Section 2.2.

ELA is a 54-amino acid preprotein processed in different isoform lengths: ELA-32, ELA-22, ELA-11 and, probably, ELA-14 and ELA-21. More precisely, as a result of proteolysis, the ELA sequence is cleaved by furin, generating ELA-11 and ELA-21 [20]. However, cleavage of the signal peptide in the N-terminus produces a 32-amino acid proprotein. ELA-32 is a mature form that becomes a biologically active molecule upon binding to APJ, similar to other isoforms [20]. Although putative furin cleavage sites were predicted to generate the other shorter peptides previously cited [27,28], the detection of a small number of them still needs to be proven in vivo.

Further research is still necessary to identify preponderant ELA and apelin isoforms and the mechanisms regulating their production, especially during physiological and pathological pregnancy. However, the high conservation of APJ, apelin, and ELA suggests that the apelinergic system is a key regulator of essential physiological functions [20,29].

## 2. The Apelinergic System in the Reproductive System—Pregnancy and Postpartum

### 2.1. Reproductive System

The topographical distribution of apelinergic-synthesizing neurons in rats [30] and the hypothalamic localization of apelin fibers and receptors [31] have suggested an implication of the apelinergic axis in behavior control and pituitary hormone release [32]. Its implication in reproductive regulation was further supported by the findings of Pope et al., who reported high levels of APJ mRNA and apelin binding sites in the mouse uterine endometrium and ovary [33]. In addition, the corpus luteum presented a high level of APJ expression. These observations suggest that the intraovarian apelinergic system may have an autocrine role [33].

Apelin and APJ are also present in bovine granulosa and oocytes. Apelin increases the secretion of basal and insulin-like growth factor 1 (IGF-1)-induced progesterone in bovine luteinizing granulosa cells, whereas it inhibits oocyte maturation and progesterone secretion from cumulus cells in vitro [34]. Accordingly, in a porcine model, apelin also increased the secretion of basal and IGF1- and FSH-induced progesterone and estradiol secretion, with an increased expression of both apelin and APJ with follicular growth [35]. In the human ovary, the apelinergic axis is localized through different developmental stages, including luteinized human granulosa cells, theca, oocytes, and the corona cumulus complex [36]. In cultured human luteinized granulosa cells, IGF-1 increased APJ expression, and recombinant human apelin stimulated the secretion of both basal and IGF1-induced progesterone and estradiol secretion [36]. The coherence of former data suggests that the apelinergic system, more specifically apelin, plays several roles in the hypothalamus–pituitary–gonadal axis and in the female reproductive organs, thus highlighting a crucial involvement in steroidogenesis [37].

### 2.2. Development of the Embryo

In human embryonic stem cells (hESC), ELA can potentiate the TGF-β pathway to prime hESCs toward the endoderm lineage [38]. It is abundantly secreted by undifferentiated hESCs, which do not express APJ [38], thus implying that ELA might use a secondary receptor [39]. ELA also appears to be an important endogenous growth factor in human embryos with a crucial role in maintaining the growth and self-renewal of human and mouse ESCs [38], which have a key function in maintaining genome stability. ELA facilitates hESC cell-cycle progression, as well as protein translation, and suppresses stress-induced apoptosis [38]. Accordingly, the inhibition of ELA causes decreased cell growth, cell death, and loss of pluripotency in hESC [38].

The apelinergic system has a complex spatiotemporal regulation in embryology, which needs to be fully elucidated and appears to be species-specific, making it difficult to extrapolate from animal models to human physiology. For example, Freyer et al. observed that, in contrast to zebrafish, ELA is not the first apelinergic ligand to be expressed in mice [40]. In fact, in mice, apelin is first expressed in extraembryonic visceral endoderm and the primitive streak at embryonic day (E) 6.5, whereas APELA expression is detected at E7.0 in the distal epiblast and shortly thereafter in the definitive endoderm [41]. At E8.25, APELA is expressed in extraembryonic tissues and in the chorion and at E9.5 in peripheric trophoblast cells [40]. These authors also observed APJ in the allantois and the vasculature invading the placenta at E9.5, which suggests that apelinergic signaling may function in extraembryonic and embryonic tissues, with an impact on the formation of mesoderm derivatives, such as yolk sac vasculature, hematopoietic progenitors, the chorion, and the allantois [40]. Interestingly, apelin knockout mice do not exhibit the endoderm defects [20] found in zebrafish [20,21], thus suggesting non-conserved roles in vertebrate gastrulation, which could be due to species-specific mechanisms of mesendoderm migration [40]. Apelin was also observed at the end of gastrulation during zebrafish heart development [20,42,43].

ELA is also a key factor in the process of gastrulation. Notably, knockdown of APELA in zebrafishes resulted in the reduced movement of ventral and lateral mesendodermal cells during gastrulation [21]. Indeed, during gastrulation, ELA increases cell velocity in a nondirectional manner toward progress in mesendoderm internalization [21]. Moreover, in zebrafish, it is also involved in guided cell migration by driving angioblast migration to the midline in dorsal aorta formation [44]. In embryo development, the ELA/APJ pathway is also implicated in skeletal development, bone formation, and bone homeostasis [45].

By contrast, ELA is essential for the proper differentiation of endodermal precursors that are known to be crucial for guiding the overlying cardiac progenitors to the heart-forming region [20]. The presence in zebrafish embryos of the grinch mutation, localized in the APLNR zebrafish ortholog, often results in the complete absence of cardiomyocytes, thus highlighting the critical role played by APLNR in myocardial development [46]. Indeed, APLNR knockdown 1-cell embryos and APLNR-deficient mice also show higher lethality due to cardiovascular abnormalities [4,5,43,46,47]. Moreover, later cardiovascular defects in adulthood were observed in most surviving mice embryos [4,5].

Paskaradevan et al. [47] demonstrated a novel mechanism for APLNR signaling in the establishment of a niche required for the proper development of zebrafish myocardial progenitor cells via the activation of Gata5/Smarcd3. However, despite the fact that apelin^−/−^ mutants exhibited cardiac developmental defects, apelin^-/-^ zebrafish [43] and mice [4,48] remained viable and fertile, suggesting that another APJ ligand other than apelin could be involved in embryonic development, i.e., ELA. The implication of ELA was later confirmed as its loss of function in zebrafish [20], and the mouse [24] model produced similar results to APJ deletion, i.e., partial embryonic lethality and cardiovascular defects. Indeed, Chng et al. observed that the loss of ELA in zebrafish embryos caused the development of a rudimentary heart or no heart at all [20]. The authors proposed a zebrafish model in which APLNR is required to fine-tune nodal output, acting as a specific rheostat for the Nodal/TGFβ pathway during the earliest stages of cardiogenesis [49]. In mice, ELA deficiency inhibits embryo blood vessel remodeling and suppresses the angiogenic sprouting of vitelline vessels, dorsal aorta, and outflow-tract and inter-somitic vessels [50]. Moreover, ELA deficiency further causes angiogenesis defects in the mouse embryo through the promotion of the expression of the endothelial cell-specific molecule 1 (ESM1) gene [50]. Likewise, in human embryos, ELA has been proposed as an endogenous secreted growth factor for hESCs that activates the TGF-β pathway to promote vasculogenesis [51].

Globally, the Elabela/APJ axis induces cardiogenesis, vasculogenesis, and bone formation during embryonic development. Furthermore, in adults, it also enhances cardiac contractility, promotes vasodilatory effects, mediates fluid homeostasis, and reduces food intake. In addition, the apelin/APJ axis is involved in embryonic vascular, ocular, and heart development [52]. Apelin has actions on blood pressure [53,54] and vasodilatation, and it has a stimulatory effect on endothelial cell proliferation that may be involved in blood vessel diameter during angiogenesis [55,56]. Of note, these cardiovascular effects of the apelinergic system in adults have not yet been studied during pregnancy.

### 2.3. The Apelinergic System in Placenta

In zebrafish, APELA is first expressed in trophoblasts and is robustly upregulated after allantoic fusion, which occurs at an early phase of placental vascular development [24]. After E10.5, ELA becomes restricted to the syncytiotrophoblasts (STBs) juxtaposed to APJ-expressing fetal endothelial cells, suggesting a paracrine mode of action [24].

Georgiadou et al. observed that, in first trimester human placentas, both ELA and apelin were expressed in villous cytotrophoblasts (CTBs), in STBs, and in distal column extravillous CTBs (EVTs), while APJ was expressed in villous CTBs and distal column EVTs, but not in STBs [57]. In addition, they observed strong ELA expression in stromal cells of term placentas [57] in some samples, indicating that trophoblasts are not the only source of placental ELA and that stromal cells might also play a functional role. However, in contradiction to this study, Inusuka et al. observed strong APJ expression on the cellular membranes of first trimester STB, whereas weaker expression was detected in villous CTBs and EVTs [58]. In addition, APJ location varies throughout human gestation. At the beginning of pregnancy, it is mainly located on the cellular membranes of STB and EVTs, and in the second and third trimesters, its presence is more pronounced in the cytoplasm of STBs [58].

The expression of apelin was also observed in the cytoplasm of the blood capillaries, the endothelium, and the placental arteries in term placentas [59]. The apelinergic system might therefore play a role in placental development, such as cell differentiation, proliferation, apoptosis, and invasion (Figure 1).

#### 2.3.1. Differentiation

The development of the placenta depends on the coordination of the proliferation and differentiation of trophoblast cells [60,61]. Each differentiation stage may be related to impaired placental development and cause placental-related pregnancy complications, highlighting the central role of differentiation in their pathogenesis [62]. ELA plays a key role in the regulation of the differentiation stage of human EVTs, including transition from a proliferative to an invasive phenotype [57]. Abnormal EVT differentiation leads to impaired invasion into the decidua by interstitial EVTs and the altered remodeling of spiral arteries by endovascular EVTs. The failure of the physiological transformation of spiral arteries has a role in preeclampsia (PE).

#### 2.3.2. Cell Cycle and Proliferation

The human trophoblast cells must exit the cell cycle in order to differentiate and fuse to form multinucleate STBs [63]. Studies have shown that the depletion of a cell cycle inhibitor (p21) could lead to the reduced expression of fusion-related genes, which adversely affects the fusion capability of trophoblastic cells [64]. Increasing evidence emphasizes the major roles of cell cycle regulators in trophoblast cell division and differentiation [65]. Several cell cycle regulators are expressed in human placenta, with distinct and dynamic expression levels [66]. Apelin-13 treatment alters cyclin expression by particularly stimulating the expression of cyclins D and E and thus the cell cycle progression in both JEG-3 and BeWo cells [59]. It has also been demonstrated that apelin-13 promotes JEG-3 proliferation via APJ and the extracellular signal-regulated kinases (ERK)1 and 2, the signal transducer and activator of transcription 3 (STAT3), and the adenosine monophosphate-activated protein kinase alpha (AMPKα) signaling pathways [59] (Figure 2). Similarly, Ma et al. recently found that ELA promoted the proliferation of BeWo cells [67].

#### 2.3.3. Cell Survival

ELA and apelin can also exert anti-apoptotic effects on BeWo cells by the activation of the PI3K-Akt pathway [67,69] (Figure 2). The apelin/APJ system increases the expression of pro-survival and decreased proapoptotic factors on mRNA and protein levels in both BeWo cells and villous explants [69]. Ferroptosis, a programmed cell death caused by iron-dependent peroxidation of lipids, might be rescued by ELAs by disrupting ferritinophagy and increasing ferritin heavy chain (FTH1) in HTR-8/SVneo cells. Interestingly, some authors report an increased grade of ferroptosis accompanied by a downregulation of the expression of ELA in PE placentas and further confirm an increased grade of ferroptosis together with a downregulation of ELA in PE-like model mouse placentas, thus providing new insights into the mechanism and therapeutic targets of PE [70].

#### 2.3.4. Trophoblastic Invasion

Abnormal EVT invasion into the decidua led to an alteration of spiral artery remodeling by endovascular EVTs and, ultimately, to utero–placental insufficiency. The addition of ELA in the culture medium of the choriocarcinoma cell line JAR was reported to increase their invasiveness in transwell invasion assays [24]. It has also been shown that the treatment of HTR-8/SVneo with apelin or ELA also increased their invasiveness [57] and is dependent on APJ [71]. In addition, ELA induces the invasion and migration of HTR-8/SVneo cells through the phosphatidylinositol-3-kinase/protein kinase B (PI3K/Akt) pathway [72] (Figure 2).

#### 2.3.5. Placental Hormone Secretion

The apelinergic system might be implicated in the production and secretion of placental hormones [73], which is probably the reason why they vary through pregnancy [74]. Apelin could decrease the secretion of protein hormones through the protein kinase A (PKA) and extracellular signal-regulated kinases (ERK1/2) signaling pathways (Figure 2) [68].

### 2.4. Labor

Apelin has been shown to inhibit human uterine contractility in vitro [75], suggesting its potential role in parturition. In rats, apelin levels were increased at the end of pregnancy and induced myometrium contractions, with their frequency and amplitude depending on its concentration. This effect does not occur with the PKC inhibitor, indicating that the PKC pathway might be implicated in its mechanism of action [76]. By contrast, an in vitro study showed that apelin suppresses both spontaneous and oxytocin-induced contractions in human myometrial fibers [75]. These contradictory results may be explained by the intracellular balance between vascular dilatation and the smooth-muscle contraction mechanisms of the apelinergic system, as well as the impact of species diversity and reagent concentrations [37].

Higher concentrations of apelin have been found in pregnant women with obesity during pregnancy, which could explain their decreased myometrial contractility, potentially due to the inhibition of the myometrial RhoA/ROCK (RhoA kinase) pathway [77]. Women with obesity have a higher frequency of cesarean sections compared to non-obese women, which is associated with an altered myometrial function that leads to a lower frequency and potency of contractions. The association of apelin and lower uterine contractility in pregnant women with obesity deserves further evaluation. Regarding ELA, neither its expression in the uterus nor its role in myometrium contractility has yet been reported.

### 2.5. The Apelinergic System and Postpartum/Breastfeeding

Apelin is abundant in breastmilk [78,79] and its level increases with long- and short-term overnutrition, possibly via maternal hyperinsulinemia and the transcriptional upregulation of apelin expression in the myoepithelial cells of the mammary gland [80]. Interestingly, the apelin level is lower in the breast milk of lactating women who have gestational diabetes [79]. At present, little is known regarding the mRNA or protein expression of APELA and ELA in the mammary gland in any mammalian species.

## 3. Placenta-Related Complications

The apelinergic system has a central role in early placentation. Early placentation dysfunction is a known trigger mechanism for placenta-related pregnancy complications. We review here current knowledge on the possible implication of the apelinergic system in several complications.

### 3.1. Preeclampsia (PE)

PE is a hypertensive disorder with multiple organ involvement. It affects 5% to 8% of all pregnancies [81] and remains the leading cause of fetal and maternal morbidity and mortality. PE and related disorders cause 14% of maternal deaths each year globally [82]. However, authors suggest that the addition of angiogenic markers to the conventional diagnostic criteria would improve the detection rate of both maternal and perinatal adverse outcomes [83]. In mice, ELA deficiency leads to hallmarks of PE such as hypertension, proteinuria, glomerular endothelial cell hyperplasia, and low birthweight (i.e., intrauterine growth restriction [IUGR]) [24], making ELA-deficient animals a suitable model for the study of PE, as well as the involvement of ELA in the pathogenesis of PE [84].

ELA deficiency in mice causes placental dysfunction characterized by a thin labyrinth, poor angiogenesis, increased apoptosis, decreased proliferation, and delayed STB differentiation [24]. In addition, circulating ELA levels correlate with the severity of maternal proteinuria and kidney damage. Interestingly, the infusion of exogenous ELA normalizes hypertension and proteinuria in ELA-deficient pregnant mice [24], suggesting that circulating ELA participates in maternal cardiovascular and renal adaptations to pregnancy independently of other well-known PE angiogenic factors (soluble fms-like tyrosine kinase-1 (sFlt-1)/placental growth factor [sFlt1/PlGF]) [24]. Moreover, Ma et al. showed that ELA significantly reversed NG-nitro-l-arginine methyl ester (L-NAME)-induced hypertension in mice, reversed the condition of maternal blood sinuses narrowing (in the placental labyrinth zone), and regulated the expression of mouse placental apoptosis factors [67]. L-NAME is a nitric oxide synthase inhibitor that disrupts uterine spiral artery remodeling in pregnant animals and increases placental vasoconstriction and vascular reactivity, and it thus decreases blood flow, leading to placental ischemia [85,86,87]. Treating pregnant rodents in their second and third trimesters with L-NAME results in hypertension, proteinuria, renal damage, IUGR, and thrombocytopenia [88,89,90].

In humans, ELA data are highly contradictory. At the protein level, translational studies do not support the hypothesis that human PE is characterized by an early deficiency in circulating ELA levels. There is no association between circulating ELA-32 in maternal blood and preterm PE [91,92]. By contrast, placental and circulating ELA-32 have been found to be elevated in two studies including women with late-onset PE [92,93] and decreased in another study by Zhou et al. [71]. In addition, Georgiadou et al. found significantly lower levels of circulating ELA in women with a normal body mass index (BMI) who later developed late-onset PE compared to women with uncomplicated pregnancies, while levels in early-onset PE did not reach statistical significance [57]. The authors suggested that ELA could not be used as a first trimester PE screening biomarker due to the large variability and dependence of ELA levels on BMI. Indeed, the study by Zhou et al. included women with a mean BMI < 25, while the study by Panaitescu et al. included a majority of women with a BMI > 25.

Another study reporting on the screening of APELA variants in PE women versus controls concluded that two rare variants were found only in PE cases, suggesting that women who express these rare variants might have a reduced transcription of the protein, which could result in an increased risk of PE [94]. Thus, the apelinergic system could be impaired in a very specific subset of women with PE, and future research should focus on their identification. More studies are also needed to identify whether specific ELA isoforms are dysregulated before the diagnosis of PE [95], but specific enzyme-linked immunosorbent assay (ELISA) tests would be required. To date, only two different ELISA tests are available for this purpose [96]. Pritchard et al. [31] and Panaitescu et al. [32] used the most frequently used one, the kit from Peninsula Laboratories (Peninsula Laboratories International, Inc., BMA Biomedicals, Augst, Switzerland), which is said to react for ELA-32, whereas Villie et al. [53] chose the kit from Creative Diagnostics (Shirley, NY, USA), which cross-reacts with human ELA-21 and ELA-32, and is more expensive.

Regarding apelin, Hamza et al. found that PE-induced rats (L-NAME) showed significantly decreased apelin serum levels [97]. Moreover, they observed significantly increased blood pressure and urine proteins. These parameters negatively correlated with the serum apelin level, and exogenous apelin-13 administration significantly improved them, together with an improvement in the placental histoarchitecture [97]. Accordingly, in reduced uterine perfusion pressure PE-induced rats, Wang et al. observed that apelin-13 treatment significantly improved the symptoms of PE, suggesting that apelin may be a potential target for treating PE [39].

There are also contradictions between the clinical results regarding the expression of apelin and its circulating levels and correlation with PE. Several studies have shown that the serum apelin level was increased in PE [98,98,99], whereas other studies showed decreased apelin mRNA and proteins in PE placentas [58] or maternal blood [100], serum [101,102], or plasma [103] levels compared with normotensive pregnancies [58,71,100,101,103,104]. However, most authors have assessed total apelin levels using the same Apelin-36 ELISA kit from Phoenix Pharmaceuticals (Burlingame, CA, USA) [96], which targets the last 12 amino acids of all isoforms and thus is said to cross-react with apelin-12, apelin-13 and apelin-36. Nevertheless, two teams also studied apelin-13 and also found decreased serum levels [104,105]. On the other hand, a recent review determined that, although there was a high heterogeneity within available studies, there was no difference in circulating maternal apelin levels between the two patient groups [106]. Nevertheless, it was observed that patients with PE had a higher BMI and lower gestational age and birthweight at delivery. When performing a subgroup analysis, PE women with a higher BMI had significantly lower apelin levels, whereas there was no significant apelin difference depending on PE severity.

Data about the apelinergic system levels in newborns are still critically lacking. However, it was demonstrated that ELA and apelin levels were decreased in newborns’ venous-arterial cord blood in women with PE and severe PE compared with healthy pregnant women [100].

### 3.2. Intrauterine Growth Restriction (IUGR)

IUGR, also called fetal growth restriction, is defined as the failure of the fetus to reach its genetically established growth potential [107,108] and is diagnosed in approximately 10% of pregnancies [109]. Malamitsi-Puchner et al. found the presence of markedly high concentrations of apelin in umbilical plasma samples, which suggests a potential role for this peptide in intrauterine growth [110]. Subsequently, it was observed that apelin levels were decreased in IUGR serum and placenta staining [111] compared to uncomplicated pregnancies or to pregnancies complicated by PE, but the study sample was too small (four cases of IUGR) to reach any conclusion. Apelin is known to stimulate proliferation and inhibit apoptosis in mouse and human osteoblasts [112], which could be a potential mechanism linking apelin and fetal growth.

As mentioned previously, ELA levels were correlated with birthweights in mice [24]. In humans, ELA serum levels have been found to be lower in cases of IUGR in one study [113] but higher in another [114]. These contradictory results might be explained by different IUGR inclusion criteria (estimated fetal weight below the third percentile in the study by Berham et al. and fetal abdomen circumference measurement below the 10th percentile in the study by Yener et al.) and different gestational ages at sample collection (at approximately 30 weeks and at delivery date for Berham, and at approximately 36 weeks for Yener). In addition, Berham et al. excluded hypertensive patients, but Yener et al. did not.

### 3.3. Gestational Diabetes Mellitus (GDM)

Apelin is known to play a role in blood glucose metabolism [56]. Two studies have shown an increase in the apelin serum level of GDM pregnant women [74,115], whereas other studies reported either decreased concentrations [115,116,117] or an absence of any difference [118,119,120,121]. Other authors studied specifically the second and third trimesters of pregnancy and found that ELA serum levels were decreased in GDM, whereas apelin serum levels increased [74]. Dasgupta et al. reported that apelin expression in GDM placentas was significantly reduced compared with matched controls [122]. Moreover, GDM mice treated with apelin showed a significant improvement in inflammatory cytokines, oxidative stress in the placenta, and glucose and lipid metabolism [123]. This suggests that the apelinergic system pathway is a promising target for the development of prophylactic and therapeutic agents for GDM in the future. However, the data are still inconsistent and more studies are required.

### 3.4. Miscarriage

Spontaneous abortions are multifactorial, but apart from genetic causes, a placental implication is plausible [124]. Placental histological changes have been reported in this field, but also delays in trophoblast development, impairment in villous vasculogenesis–angiogenesis [125], and insufficient syncytialization [126]. ELA-like APLNR null mice [40] and zebrafish [20] have reduced survival, probably mainly due to heart development and placental defects, but little is known about the direct influence of the apelinergic system on spontaneous abortion. To our knowledge, there is only one publication demonstrating an association of lower maternal ELA levels with spontaneous abortion [127].

## 4. From Research to the Clinical Setting: Challenges and Limitations

Studies have reported an alteration of circulating and placental ELA and apelin levels in pathologies of pregnancy such as PE and GDM, suggesting that these peptides could be used as biomarkers (Table 1). However, the results of these studies vary significantly (Table 1). First, obtaining specific proper dosages of apelinergic ligands is challenging. Apelin and ELA protein levels have mainly been evaluated using commercial antibody-based immunoassays against various synthetic peptide fragments [73,128,129]. These assays are probably not isoform-specific [130]; this may contribute to the reported large plasma concentrations ranges observed. High-performance liquid chromatography combined with radioimmunoassay detection has also been used, with a confirmed relative specificity for different apelin isoforms [131,132,133], but only a small number of authors have successfully detected apelin in vivo [130,134,135].

Second, the variability of results using the same commercial antibody-based immunoassays suggests critical variances in the specificity of ELA tests from different manufacturers. Indeed, using the same ELISA kit, Pritchard et al. [91] and Panaitescu et al. [92] found different ELA concentrations in samples collected at term (~30 pg/mL and 5 ng/mL, respectively).

A third limitation of the evaluation of the apelinergic system is the lack of guidelines and recommendations for sample handling procedures, such as sample extraction, to ensure the correct use of the ELISA kit. Georgiadou et al. [128] reviewed eight studies reporting data on circulating ELA in human plasma or serum. These studies used ELISA kits from three different companies (Phoenix Pharmaceuticals, Peninsula Laboratories International, and Creative Diagnostics) all recommending sample peptide extraction by means of high performance liquid chromatography. In two studies [25,141], it appears that sample extraction was not performed. In the other studies, the procedures for sample extraction were unclear, making their results difficult to interpret. Interestingly, when the authors indicated that peptide extraction was performed, the results obtained by the commercial kit mirrored the levels obtained by the custom ELISA used by Georgiadou et al., whereas when peptide extraction was not performed, the ELA levels no longer showed their typical large inter-individual variations. In addition, all isoforms have a short half-life, probably of only a few minutes [28], thus implying a rapid and transient response to homeostatic changes and differences in potency [142] and signaling pathways, resulting in difficulties in establishing their correct dosage and in their subsequent use as a therapeutic agent. Thus, further studies are needed to establish guidelines for sample handling and for the measurement and conservation of apelinergic system isoforms.

Another limitation is the variation in the population’s intrinsic characteristics. Apelin varies according to body mass index [143,144], patient age [145], and the presence of diabetes [146,147], or in cases of an inflammatory condition, such as psoriasis [148]. Variations linked to thyroid disorders have also been evaluated [149,150] but are not conclusive. Moreover, apelinemia could decrease during the second half of pregnancy [151,152]. Additionally, as the apelin gene is located on the X chromosome, analyses are sometimes conducted in a sex-specific manner. For example, specific apelin genotypes were associated with lower high-density lipoprotein cholesterol in Iranian women without a metabolic syndrome [153]. Likewise, gender-specific associations between apelin/APJ gene polymorphisms have been highlighted in humans [154]. In the APLNR knockout mouse model, sex-specific effects on conditioned fear responses were observed [155].

ELA levels correlate with age, BMI, heart rate, BNP levels, and left atrial dimension [156]. ELA levels also appears to be higher during pregnancy [24,25] and linked to gestational age [128]. In 50% of women in the resistance to aspirin during and after pregnancy (RADAR) cohort, ELA could not be detected throughout pregnancy and the postpartum period [57]. However, Panaiteiscu et al. were able to detect ELA in the plasma of women during the second and third trimesters [92]. Of note, ELA levels were highly variable between women and did not particularly change during pregnancy, although a slightly higher level was observed in the first trimester [57]. Finally, both the apelin and the ELA peptide levels might be altered in some other pathologies of pregnancy such as GDM, which is an issue that must be considered.

Preclinical studies have also evaluated these peptides for the treatment of PE. Regardless of the rat PE models used, the administration of apelin or ELA peptides significantly improved the symptoms of PE (Table 2). However, these studies were only conducted in rats with PE, and these results should be confirmed in other animal models. Indeed, the spatiotemporal regulation of the apelinergic system in embryology appears to be species-specific [40,41], making it difficult to extrapolate from animal models to human physiology.

## 5. Conclusions

The apelinergic system in the reproductive field plays a central role in both physiological pregnancy and placenta-related complications in pregnancy. More specifically, apelin has a crucial role in steroidogenesis and in the metabolic regulation of the ovary. ELA also has a potential role in the onset of PE. These data suggest that the apelinergic system is a promising research field with a translational potential related to therapeutic interventions in pregnancy. Further studies investigating the role of the apelinergic system in the development of early pregnancy development and complications remain necessary to fully understand its role and its potential in the development of therapeutic strategies.

## Figures and Tables

**Figure 1 ijms-24-08014-f001:**
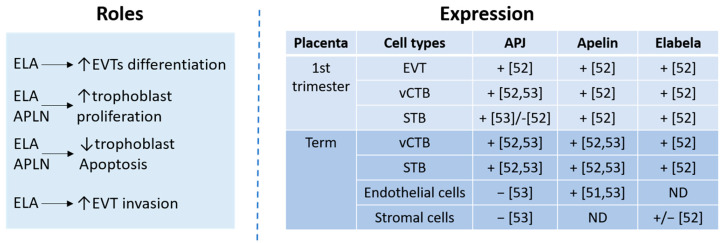
Apelinergic system expression and roles in placenta. ELA: Elabela; EVT: extravillous trophoblast.

**Figure 2 ijms-24-08014-f002:**
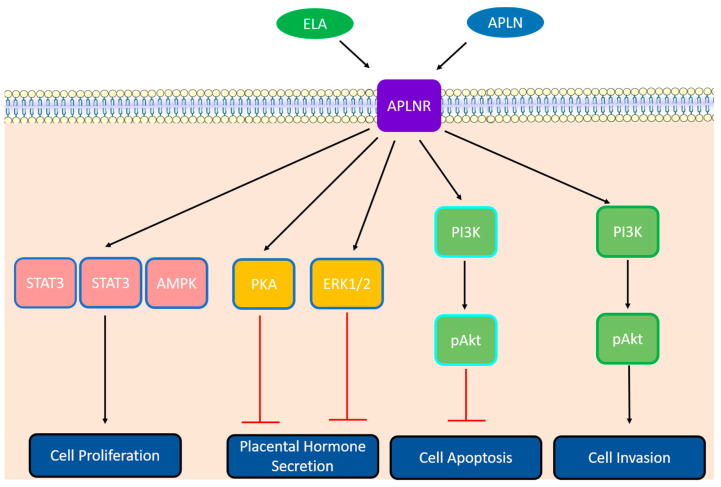
Activation of different signaling pathways through ELA (framed in green), apelin (framed in green), or both (framed in cyan) in the binding of APJ in human trophoblast [59,68,69]. ELA: Elabela; APJ: apelin peptide jejunum; AMPK: adenosine monophosphate-activated protein kinase; ERK1/2: extracellular signal-activated kinase 1/2; PKA: protein kinase A; PI3K: phosphatidylinositol 3-kinases.

**Table 1 ijms-24-08014-t001:** Altered APJ, apelin, and ELA levels in human pregnancy disorders compared to a physiological pregnancy during pregnancy and in the postpartum period.

Variable	Tissue/Fluid	Expression	Methods	Pathologies	Reference
**Pregnancy**					
**Molecule: APLNR**					
delivery	placenta	mRNA, protein	Real-time PCR, IHC	Late onset PE (↘)	[71]
delivery	placenta	protein	IHC	PE (≈)	[58]
delivery	placenta	protein	IHC	PE (↗)	[136]
delivery	placenta	mRNA	RT-qPCR	Maternal obesity (≈)	[137]
delivery	placenta, adipose tissue	mRNA	Real-time PCR	GDM (≈)	[138]
**Molecule: APLN**					
delivery	serum, cord blood	protein	ELISA	GDM (≈in cord blood, ↗ in serum)	[115]
24–28 WG	serum	protein	ELISA, EIA	GDM (↘)	[117]
2nd and 3rd trimester	serum	protein	ELISA	GDM (↗only in the 2nd trimester )	[74]
delivery	serum	protein	ELISA	GDM (≈)	[118]
delivery	maternal and cord blood	protein	ELISA	GDM (↘ in cord blood, ≈ in maternal blood )	[119]
24–32 WG and delivery	plasma, adipose tissue, placenta	protein	ELISA, Real-time PCR	GDM (≈mRNA and circulating level)	[138]
delivery	placenta	protein	IHC	GDM (↘)	[122]
2nd trimester	serum	protein	ELISA	GDM (↗)	[139]
20–34 WG and delivery	serum, placenta	protein, mRNA	ELISA, IHC, RT-PCR	Preterm ≈, IUGR and PE (↘ prot, ≈mRNA)	[111]
delivery	placenta	protein	RIA	PE (↘)	[140]
delivery	plasma	protein	ELISA	PE (↘)	[103]
time of diagnosis	serum	protein	EIA	PE (↗)	[98]
time of diagnosis	serum	protein	EIA	PE (↘)	[101]
delivery	maternal and cord blood	protein	ELISA	PE (↘ in both maternal and cord blood)	[100]
delivery	placenta	protein	IHC	PE (↘)	[122]
delivery	serum	protein	ELISA	PE (↘)	[104]
delivery	placenta, serum	protein, mRNA	ELISA, IHC, WB, RT-qPCR	PE (↘ in placenta, ↗ in maternal circulation)	[58]
delivery	serum	protein	ELISA	PE (↘)	[105]
delivery	placenta	protein	IHC	PE (↗)	[136]
time of diagnosis	serum	protein	ELISA	PE (↗)	[99]
delivery	plasma, cord blood, placenta	protein, mRNA	ELISA, RT-qPCR	Maternal obesity (≈mRNA and plasma, ↘ in cord blood)	[137]
**Molecule: ELA**					
time of diagnosis	serum	protein	ELISA	IUGR (↘)	[113]
delivery	serum	protein	ELISA	IUGR (↗)	[114]
delivery	maternal and cord blood	protein	ELISA	PE (↘ in both maternal and cord blood)	[100]
delivery	plasma	protein	ELISA	Early-onset PE (≈), late-onset PE (↗)	[92]
delivery	serum, urine, placenta	protein, mRNA	ELISA, IHC, Real-time PCR	Late-onset PE (circulating and placental level ↘)	[71]
delivery	plasma, placenta	protein, mRNA	ELISA, RNA sequencing	PE (≈ placental mRNA and circulating protein)	[91]
1st trimester	serum	protein	ELISA	GH/PE (≈)	[67]
2nd trimester	serum	protein	ELISA	GDM (↗)	[139]
2nd and 3rd trimester	serum	protein	ELISA	GDM (↘ during second trimester)	[74]
delivery	plasma, cord blood, placenta	protein, mRNA	ELISA, RT-qPCR	Maternal obesity (≈mRNA and protein)	[137]
time of diagnosis	serum	protein	ELISA	MA (↘)	[127]
**Post-partum**					
**Molecule: APLN**					
	serum, colostrum and mature milk	protein	ELISA	GDM (↘ in colostrum and milk)	[79]
	plasma	protein	ELISA	GDM (↘)	[116]
	plasma, breast milk	protein	ELISA	Obesity (↗with BMI)	[80]

PCR: polymerase chain reaction; IHC: immunohistochemistry; PE: preeclampsia; GDM: gestational diabetes mellitus; IUGR: intrauterine growth restriction; EIA: enzyme immunoassay; ELISA: enzyme-linked immunosorbent assay; MA: missed abortion.

**Table 2 ijms-24-08014-t002:** Effect of administration of apelin and ELA peptides during pregnancy pathologies [39,97,140,157,158].

Model	Peptide	Output	Reference
Obesogenic diet mice	Apelin	Improvement in placental function of obese dams	[157]
L-NAME-induced PE in rats	Apelin-13	Reno-protective effects	[97]
PE rat model (TGA-PE)	Pyr- Apelin-13	Improvement in hemodynamic response and renal injury without fetal toxicity	[140]
PE rat model by reduced uterine perfusion pressure	Apelin	Amelioration of PE symptoms	[39]
L-NAME-induced PE in rats	ELA	Reversion of the phenotypes of L-NAME-induced PE	[158]

PE: preeclampsia; L-NAME: L-nitro-arginine methyl ester; ELA: Elabela. Many technical challenges need to be overcome before these can be used in clinical settings.

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
