# Peer review of "The Apelinergic System in Pregnancy"

_ijms, 2023, doi:10.3390/ijms24098014_

Round 1
Reviewer 1 Report
Authors should thoroughly read the literature on the topic before starting the writing of the manuscript.
Dawid et al., has previously published a paper on a similar subject:
https://pubmed.ncbi.nlm.nih.gov/35011661/
I believe that the article is not innovative and does not bring anything new to knowledge.
Author Response
Re: Manuscript ID: ijms-2307016 “The Apelinergic System in Pregnancy” by Pécheux O, et al
(Rev 1)
Reviewer #1
Reviewer 1: Authors should thoroughly read the literature on the topic before starting the writing of the manuscript. Dawid et al., has previously published a paper on a similar subject: https://pubmed.ncbi.nlm.nih.gov/35011661/. I believe that the article is not innovative and does not bring anything new to knowledge.
Response
Thank you for the comment. Indeed, we are aware of the excellent literature review by Dawid et al, which is already cited in our manuscript (p. 8, ref 73). However, it focused on the expression and effects of Apelin, APJ and Elabela on the placenta and pathologies of pregnancy associated with impaired placentation, but it did not report on their expression and role more broadly in the reproductive system or in early embryonic development, labor or breastfeeding. In addition, in our review, we discuss the clinical interest of these peptides by determining their limitations and the challenges to be overcome.

Reviewer 2 Report
This comprehensive review summarises the current knowledge of the aplinergic system in pregnancy.
Throughout the manuscript please be clear on what species you are referring to, what stage of pregnancy, and what type of sample (eg serum, protein in placental homogenates....). As you are referring to many species and studies, this is essential for the reader to fully appreciate the findings you are discussing.
Figure 1 and 2- please reference the manuscripts you are getting this information from in the legend.
Figure 1: The right hand side of this figure with the preeclampsia is perhaps confusing/misleading as you then go on to discuss the inconsistencies in the published studies regarding the relationship between preeclampsia and these molecules. Perhaps may be useful to have a diagrammatic representation of the localization of each of the components of the system at different stages of pregnancy or a comparison between the animal models you have discussed and humans
2.3.1 - needs expanded with more specific details here
2.5 - last sentence of this paragraph is confusing. Perhaps should be something more along the lines of 'At present, little is known regarding the mRNA or protein expression of Apela and ELA in the mammary gland in any mammalian species'
Assuming Table 1 is all human data? Please specify this. Also would be good to detail at what stage of pregnancy the differences are observed that have been linked to the outcome.
So in section 4 you nicely summarize a lot of human clinical data. However, a lot of this content has been detailed in the previous section so it feels as though there is unneccessary repetition of some of the findings. Perhaps things need restructured slightly to minimize repetition and help with the flow of the manuscript.
As the apelin gene is located on the x chromosome, I wonder if there are any reports of differences between pregnancies with male vs female fetuses?
Author Response
Re: Manuscript ID: ijms-2307016 “The Apelinergic System in Pregnancy” by Pécheux O, et al
(Rev 1)
Reviewer # 2
We thank the reviewer for the constructive and pertinent comments.
Throughout the manuscript please be clear on what species you are referring to, what stage of pregnancy, and what type of sample (eg serum, protein in placental homogenates....). As you are referring to many species and studies, this is essential for the reader to fully appreciate the findings you are discussing.
Response
Thank you for this valuable remark. We have now clarified what species we are referring to, as well as the stage of pregnancy and the type of sample (see lines 36, 58, 113, 137, 145, 154, 162, 173, 225, 242, 351, 359, 388, 397, 429).
Figure 1 and 2- please reference the manuscripts you are getting this information from in the legend.
Response
As requested, we have now included the relevant references in the figure legends.
Figure 1: The right hand side of this figure with the preeclampsia is perhaps confusing/misleading as you then go on to discuss the inconsistencies in the published studies regarding the relationship between preeclampsia and these molecules. Perhaps may be useful to have a diagrammatic representation of the localization of each of the components of the system at different stages of pregnancy or a comparison between the animal models you have discussed and humans.
Response
According to your suggestion, Figure 1 has been modified. The preeclampsia part was removed and we added the localization of each of the components of the system at different stages of pregnancy.
2.3.1 - needs expanded with more specific details here.
Response
Following your comment, we have now expanded the paragraph (lines 206-209).
2.5 - last sentence of this paragraph is confusing. Perhaps should be something more along the lines of 'At present, little is known regarding the mRNA or protein expression of Apela and ELA in the mammary gland in any mammalian species'
Response
Thank you for this suggestion; we the sentence has now been changed (lines 283-284).
Assuming Table 1 is all human data? Please specify this. Also would be good to detail at what stage of pregnancy the differences are observed that have been linked to the outcome.
Response
We confirm that Table 1 is all human data (specified on line 429). We have also detailed at what stage of pregnancy the differences are observed that have been linked to the outcome.
So in section 4 you nicely summarize a lot of human clinical data. However, a lot of this content has been detailed in the previous section so it feels as though there is unneccessary repetition of some of the findings. Perhaps things need restructured slightly to minimize repetition and help with the flow of the manuscript.
Response
We agree with the reviewer and have deleted some sentences to avoid unnecessary repetitions (lines 328-329, 331-332, 502-505, 519-523).
As the apelin gene is located on the x chromosome, I wonder if there are any reports of differences between pregnancies with male vs female fetuses?
Response
We thank the reviewer for this comment. Indeed, there are several reports on differences between pregnancies with male vs female fetuses and we have now included these findings (lines 489-493).

Reviewer 3 Report
Pécheux et al. reviews current knowledge on the roles of the apelinergic system in pregnancy and its pathophysiological complications in placenta and embryo development. They provide valuable insights about the inconsistency in current data for circulating levels of AJP peptides and highlighted the challenges to translate the actors of the apelinergic system into a marker or target for therapeutic interventions in obstetrics.
Given most physiological studies about APJ system are focused on kidney and cardiovascular system, a review article about the APJ in placenta and fetus is valuable.
A few suggestions for the improvement of this review:
1) since APLNR is a GPCR and the two ligands, apelin and ELA, form different complexes with APLNR, please summarize the differences in cellular signaling pathways in the context of pregnancy.
2) The APJ system has both essential roles in maternal health and embryo cardiogenesis. More clear distinction of ELA’s effects in these two circumstances are needed.
3) Please summarize the different cleavages of apelin and ELA in the context of pregnancy related complications.
Minor points:
1) Line 60 mentioned Section 3B. However, there was no such section. Please correct.
Author Response
Re: Manuscript ID: ijms-2307016 “The Apelinergic System in Pregnancy” by Pécheux O, et al
(Rev 1)
We thank the reviewer for the constructive criticism provided.
A few suggestions for the improvement of this review:
1) since APLNR is a GPCR and the two ligands, apelin and ELA, form different complexes with APLNR, please summarize the differences in cellular signaling pathways in the context of pregnancy.
Response
Following your comment, we have added clarifications in Figure 2.
2) The APJ system has both essential roles in maternal health and embryo cardiogenesis. More clear distinction of ELA’s effects in these two circumstances are needed.
Response
As suggested, we have now included additional text to provide a more clear distinction between maternal health and embryo cardiogenesis (lines 132-170).
3) Please summarize the different cleavages of apelin and ELA in the context of pregnancy related complications.
Response
Following your comment to summarize the different cleavages of apelin and ELA in the context of pregnancy-related complications, new text has been added in lines 340-347 and 361-365.
We have also added details about the studied ELA isoforms in lines 62-66. However, to the best of our knowledge, there is no additional specific information available at present about the different cleavages in pregnancy. Thus, we have now added on lines 70-71: "… especially during physiological and pathological pregnancy".
